# Study of element concentrations in blood serum of patients receiving parenteral nutrition using total reflection X-ray fluorescence analysis

Monika Pierzak-Stępień[1], Aldona Kubala-Kukuś[2,3]*, Dariusz Banaś[2,3], Ilona Stabrawa[2,3], Jolanta Wudarczyk-Moćko[3], Karol Szary[2,3], Monika Biernacka[2], Milena Piotrowska[2], Dariusz Pasieka[2], Natalia Wojtaś[2], Andrzej Dąbrowski[2,3], Stanisław Głuszek[3,4,5]

1 Institute of Health of Sciences, Faculty of Health of Sciences, Jan Kochanowski University, Kielce, Poland, 2 Institute of Physics, Faculty of Natural Sciences, Jan Kochanowski University, Kielce, Poland, 3 Holycross Cancer Center, Kielce, Poland, 4 Hospital of the Ministry of Interior and Administration, Kielce, Poland, 5 Institute of Genetics and Animal Biotechnology, Polish Academy of Sciences, Magdalenka, Poland

* aldona.kubala-kukus@ujk.edu.pl

## Abstract

Total reflection X-ray fluorescence (TXRF) analysis was used to determine element concentrations in blood serum samples of patients receiving parenteral nutrition. The concentrations of elements were measured in the serum samples two times, namely in the first day (measurement I) of the treatment and on the seventh day (measurement II) after nutrient supplementation. For comparison purposes, also serum samples of 50 patients without nutritional disorders, admitted to a planned cholecystectomy (surgical procedure of the gall bladder removal), were analyzed and treated as the control group. Descriptive statistics of measured concentrations of P, S, Cl, K, Ca, Cr, Fe, Cu, Zn, Se, and Br were determined both for the studied and control groups. Statistical analysis of the results concentrates on verifying the type of element concentration distributions. Next, the comparison of element concentration between groups of women and men was done. Parenteral nutrition resulted in a significant increase in supplemented Se and a decrease in Cu concentrations between measurement I and II. Parametric as well as nonparametric approaches were applied to verify the hypothesis. Additionally, it was found out that for some elements differences between the studied and control groups can be observed (P, S, Ca, Fe, Cu, Zn, Se, Br). The analyses were also performed taking into account the gender of the patients. Correlation coefficients were calculated for the studied and control groups as regards the determined elements. It was observed that in all groups positive correlations were obtained between S and Ca, and for Zn and Se. In studied and control groups a negative correlation between Ca and Cr was obtained. This paper discusses also prediction of trace element concentrations following seven days of parenteral nutrition therapy, based on the knowledge of various anthropometric,

**Data availability statement:** All relevant data are within the manuscript and its Supporting Information files.

**Funding:** The functioning of the facility is supported by Polish Ministry of Education and Science (project 28/489259/SPUB/SP/2021). This work was co-financed by the Minister of Science (Poland) under the "Regional Excellence Initiative" program (project no.: RID/SP/0015/2024/01). The funders had no role in study design, data collection and analysis, decision to publish, or preparation of the manuscript.

**Competing interests:** The authors have declared that no competing interests exist.

biochemical and immunological parameters describing the patient's condition. The procedure is discussed for Br concentration.

## Introduction

Parenteral nutrition is a medical procedure aimed at maintaining or improving nutritional status of patients who, due to their health condition, are unable to meet nutritional needs through natural oral intake. Clinical nutrition is a multi-component process that includes: assessment of the patient's nutritional status, evaluation of the patient's nutritional requirements, providing an appropriate daily amount of macronutrients, micronutrients, and fluids (either intravenously or through the gastrointestinal tract, depending on the patient's condition), and ongoing monitoring of the therapy. Parenteral nutrition involves intravenous infusion of nutrient mixtures, either through peripheral veins (via a cannula) or a central vein (via a catheter). These mixtures typically contain proteins, electrolytes, energy sources, vitamins, water, and trace elements. This therapy is indicated in such conditions as: severe burns, chemotherapy or radiotherapy, coma, malabsorption syndromes, acute pancreatitis, intestinal obstruction, and severe eating disorders (e.g., anorexia, bulimia). It helps restore nutritional balance and supports recovery, with formulations tailored individually based on the clinical and physiological characteristics of the patient [1].

In parenteral nutrition, careful adjustment of nutrient intake is critical, as excess nutrient intake can cause refeeding syndrome, a potentially life-threatening condition [2]. Providing an appropriate amount of nutrients to hospitalized patients is essential for recovery. Parenteral nutrition mixtures typically contain water, electrolytes, amino acids, lipids, carbohydrates, and trace elements. Water, constituting about 50% of body weight in women and 60% in men, is crucial for the cellular structure, internal transport, thermoregulation, metabolic reactions, and waste elimination. During illness, fluid losses increase, requiring careful fluid management to avoid dehydration or overload. Proteins, formed by amino acids linked by peptide bonds, are essential nutrients. In parenteral nutrition, around 18 amino acids are delivered directly into the bloodstream to support healing and metabolic functions. Protein requirements increase in cases such as extensive burns or hemorrhages. Lipids provide energy and aid in the absorption of fat-soluble vitamins (A, D, E, K), essential for immune, nervous, and hormonal systems. They also form a key component of cell membranes. Lipids are administered as emulsions containing long-chain fatty acids and must be sterile, isotonic, apyrogenic, and within a pH range of 6.5–8.5. Carbohydrates are another major energy source, covering about 45–65% of caloric intake. Glucose is the most common carbohydrate in parenteral nutrition, though other sugars like xylitol or sorbitol may be used. Proper carbohydrate supply prevents ketosis and micronutrient deficiencies. Vitamins, trace elements, and electrolytes are indispensable for physiological functions and recovery. Essential components include electrolytes ($Na^+$, $K^+$, $Mg^{2+}$, $Ca^{2+}$, phosphates), vitamins (A, B-complex, C, D, E, K, folic acid), and trace elements (Fe, Zn, Cu, Mn, Mo, Se, Cr, F, I).

Parenteral nutrition can be delivered through two systems: All-in-One (AIO), combining all nutrients in a single infusion, or Two-in-One (TIO), commonly used in pediatric patients [3].

As regards the complexity of the parenteral nutrition process, it is important to monitor the patient's condition. In assessing the content of elements, including trace elements, diagnostics can be performed based on an analyzing elemental composition of the patient's serum. Serum is often chosen for elemental analysis because it does not contain cellular components or coagulation factors, which allows for precise and stable determination of elemental concentrations. In this analysis, the total reflection X-ray fluorescence analysis (TXRF) [4,5] can be applied, which is a powerful analytical technique widely utilized for elemental analysis in biological samples, including human serum. Due to its high sensitivity, capability for simultaneous multi-element analysis, and minimal sample preparation requirements, TXRF has gained significant recognition in biomedical research and clinical diagnostics. Monitoring of blood serum elemental composition is essential to the study the influence of environmental pollution, nutrition and/or occupational exposure on human health [6]. The studies of element concentration changes in serum samples are very often related to diagnostic process performed in various diseases or during treatment procedures [7–9]. The usefulness of the TXRF technique for measuring trace elements in serum samples has already been systematically tested and confirmed [10,11]. The TXRF elemental analysis of serum samples is usually performed as a direct analysis, without prior dilution, with the results aligning well with those obtained via other methods such as atomic absorption spectroscopy (AAS), inductively coupled plasma mass spectrometry (ICP-MS) and inductively coupled plasma optical emission spectroscopy (ICP-OES) [12].

This work focuses on the analysis of human serum samples of patients receiving parenteral nutrition from the Clinic of General, Oncological and Endocrinological Surgery, Regional Hospital in Kielce, Poland. In order to measure the concentrations of elements the samples were investigated using the TXRF technique. The serum elemental analysis was performed two times: on the first day of parenteral nutritional treatment (measurement I) and on the seventh day afterwards (measurement II). In the studies, also serum samples for the control group (50 persons) were analyzed, which made it possible to compare with the values obtained for the studied group. For both groups also anthropometric, biochemical and immunological parameters, e.g., body mass index (BMI), nutritional status, level of the C-reactive protein (CRP), albumin, total protein, leukocytes, and lymphocytes, were measured.

In the first part of the studies, the analysis concentrated on determining chromium and selenium concentrations after supplementing these elements during parenteral nutrition and on changes in bromine concentration, element rarely studied especially in conditions of parenteral nutrition [8]. In the studies presented in this article, we expand the research also towards the content of other elements determined in serum samples.

## Materials and methods

### Sample description

The presented studies include patients of the Clinic of General, Oncological and Endocrinological Surgery, Regional Hospital in Kielce, Poland. The research project was approved by the Bioethics Committee at the Jan Kochanowski University in Kielce, Poland (issue 8/2017). The recruitment period for this study started on August 30, 2017 and ended on January 7, 2019. The patients signed an informed consent to carry out scientific research on the collected biological material. The study not included minors. Nutritional assessment was performed using: a questionnaire containing a nutritional interview, a standardized screening research tool the NRS 2002 (Nutritional Risk Score), anthropometric, biochemical and immunological tests [8]. An analysis of the screening results for the assessment of the risk associated with malnutrition NRS allowed for assessment of nutritional risk and determination of indications for nutritional treatment. Each patient taking part in the study (patients receiving parenteral nutrition, patients from the control group) underwent the following anthropometric tests: measurement of body mass, measurement of body height, calculation of the BMI index (expressed using the formula: body mass (kg)/ height (m$^2$), assessment of body mass loss over the last 3–6 months, calculation of the ideal body mass, assessment of adipose tissue resources using a skinfold caliper measuring the thickness of the skinfold over

the triceps muscle of the non-dominant arm (TSF), measurement of the arm circumference measured with a tape midway between the coracoid and olecranon processes (MAC), calculation of the arm muscle circumference index (OMR indicates the protein nutritional status of the body), calculation of the waist to hip ratio (WHR). Additionally, the following biochemical and immunological tests were performed: determination of albumin concentration, total protein, total lymphocyte count, CRP, ionogram (Na, K, Mg, Cl, and P) and daily nitrogen excretion determined in urine.

The studied group of patients receiving parenteral nutrition consisted of 21 women and 27 men. Among women, the average age (in years) was 64 and the median was 66. The youngest women was 30, while the oldest was 80. Among men, the average age (in years) was 59 and median was 58. The youngest man was 30, while the oldest was 80 [8]. Patients from the studied group were fed using the *All-in-One* method, involving the use of one food bag. To patients the following nutritional mixture types were administered: Nutriflex Plus, Nutriflex Lipid Peri, Nutriflex Lipid Peri N4, Kabiven Peripheral and Smof Kabiven. Trace elements contained in the Addamel supplement and vitamins contained in the product called Cernevit were added to each of the food bags. Individual components were dosed in quantities determined individually, taking into account the patient's clinical condition, body weight and laboratory test results. Nutrition was provided through central and peripheral vessels in a continuous infusion, using MEDINA infusion pumps, over at least 16 hours daily. During nutritional therapy, each patient was subject to daily clinical monitoring, which included fluid balance, measurement of body temperature and weight, and observation of the central and/or peripheral venous catheter. In the study, the patients were supplemented with selenium and chromium. Standard chromium supplementation in the form of chromium III hexahydrate in food sacks used in intravenous nutrition in 1 mL of the drug is 5.33 µg/g which corresponds to 0.02 µmol. Standard supplementation of selenium in the form of anhydrous sodium selenite in 1 mL of the drug is 6.90 µg/g, which corresponds to 0.04 µmol.

Assessment of the patients' health status, anthropometric, biochemical and immunological tests and, above all, elemental analysis of serum samples using total reflection X-ray fluorescence method was performed twice, on the first day (measurement I, before starting parenteral nutrition) and on the seventh day (measurement II) of parenteral nutrition treatment. Blood samples were collected in the morning, before the feeding bag was connected, in the surgical room of the Department of Surgery.

In the study, for comparison purposes, also the group of patients not receiving parenteral nutrition, was included. This control group consisted of 50 patients (35 women and 15 men) scheduled for planned gallbladder removal surgery (cholecystectomy), aged 30–80. Criteria for including patients within the control group were: age (from 30 to 80 years), patients admitted for planned cholecystectomy, without symptoms of cholecystitis, normal nutritional status and obtaining informed consent from the patient to participate in the study. Among women, the average age (in years) was 53 and median was 50. The youngest woman was 30, while the oldest was 80. Among men, the average age (in years) was 54 and median was 60. The youngest man was 30, while the oldest was 80 [8]. For the patients in the control group, all anthropometric, biochemical and immunological parameters and elemental analysis of the serum were measured once, before cholecystectomy.

Laboratory analyses of biochemical and immunological parameters, both for the studied and control groups, were carried out in the Diagnostic Laboratory of the Regional Hospital in Kielce. Venous blood was centrifuged for 15 minutes (4000 rpm). Immediately after centrifugation, blood serum was divided into smaller portions and stored in the laboratory in Ependorf tubes of 1.5 mL volume, at −60˚C. Collected serum samples were systematically forwarded to the Institute of Physics of the Faculty of Natural Sciences of Jan Kochanowski University for elemental analysis using the TXRF method [8].

## TXRF elemental analysis of serum samples

Total reflection X-ray fluorescence measurements were performed with the S2 Picofox spectrometer (Bruker) equipped with the 30W Mo-anode X-ray tube operated at a voltage of level 50 kV with the electron current of 0.6 mA. The primary X-ray

beam from the X-ray tube, monochromatized using the Ni/C multilayer monochromator to 17.5 keV, was directed onto the studied sample below the critical angle of total reflection. Fluorescence X-rays from the samples were detected with the XFlash silicon drift detector having energy resolution about 160 eV for Mn-Kα line. The measurements were performed in air. The qualitative analysis of the X-ray spectrum and the quantitative analysis of the sample elemental composition were done using spectrometer software (SPECTRA 7) based on the spectrometer calibration. The instrument calibration is performed each day at the beginning of the measurements. The TXRF analysis validation was performed by investigating reference control serum solution at the beginning of the studied sample measurement [8]. The obtained accuracy for the TXRF analysis of reference serum samples was on the level from 1% to 9%, depending on the analyzed element. Measurement repeatability, checked using a reference serum sample, was 3–5%. Detailed information is presented in the article [8]. Detection limits for the TXRF serum analysis were estimated from about 0.008 mg/L (element atomic number 32–35) to 3 mg/L (for P) for elements analyzed by the K-series of characteristic radiation and from about 0.02 mg/L (atomic number about 60) to 0.05 mg/L (atomic number 70–80) for elements analyzed by the L-series. The uncertainty of the determined concentration depends on the element and its content in the sample. For concentration levels higher than 0.05 mg/L, the uncertainty was on the level of 5–10%, while for concentration levels less than 0.05 mg/L, the uncertainty reached values up to 50% (additional information in [8]). Information about the accuracy, the repeatability, the detection limits and the experimental uncertainties of TXRF analysis of serum control samples are summarized in the Table 1.

Serum samples of the patients from the studied and control groups were prepared according to the following procedure: 0.8 mL of a sample was mixed with 0.05 mL of Ga (100 µg/g) used as an internal standard. Next, 5 µl of solution was deposited on a quartz sample carrier and dried on a heating plate at a temperature of 40˚C. The dry residuum was next analyzed using the TXRF technique with the measurement time of 1800 s. For each sample the duplicate measurements were performed. The registered X-ray spectra of the irradiated serum samples enabled simultaneous multi-element detecting the following elements: (P), sulfur (S), chlorine (Cl), potassium (K), calcium (Ca), chromium (Cr), manganese (Mn), iron (Fe), nickel (Ni), copper (Cu), zinc (Zn), selenium (Se), bromine (Br), rubidium (Rb), strontium (Sr), and lead (Pb). The observed variability of duplicate measurements depends on the given element and its concentration in the sample and is on the level from about 0.1% to 20%.

## Results

### Elemental contents in serum samples

In our previous paper [8], concentration changes of chromium, selenium and bromine in the serum of people receiving parenteral nutrition were discussed. Chromium and selenium were supplemented, while bromine naturally occurred in the serum. The statistical analysis of the concentrations of these elements was aimed at calculating the values of descriptive statistics and verifying the hypothesis related to the comparison of element concentrations obtained in measurement I and measurement II (studied group) and the control group. The main observation of these studies was a statistically significant increase in selenium content in serum sample of patients with parenteral nutrition after treatment applied. In the case of

**Table 1. Accuracy, repeatability, detection limits and experimental uncertainties of TXRF analysis of serum control samples.**

| Accuracy [%] | Repeatability [%] | Detection limit [mg/L] | Uncertainties [%] C – element concentrations |
|---|---|---|---|
| 1-9 | 3-5 | 15-39 atomic number range: from 0.008 mg/L to 3 mg/L 60-80 atomic number range: from 0.02 mg/L to 0.05 mg/L | 5–10% (for C > 0.05 mg/L) up to 50% (for C < 0.05 mg/L) |

chromium and bromine concentration, there were no statistically significant changes in concentration, which was statistically less than the reference value [8].

Among the elements (P, S, Cl, K, Ca, Cr, Mn, Fe, Ni, Cu, Zn, Se, Br, Rb, Sr, and Pb) determined in the serum of patients, it was observed that for Mn, Ni, Rb, Sr and Pb the content of the elements was at the limit of detection of the TXRF technique [13]. This means that for some samples the obtained results of the element content were known precisely, while for the remaining ones only the detection limit was estimated [13]. Such data are an example of the random left-censored data, when the result of the observation or measurement is not known precisely, and its value is limited to a certain range [13]. The censoring level, defined as the ratio of the number of incomplete measurements to the total number of all measurements, for elements: Ni, Rb, Sr, and Pb, was in the range from 4% (Pb, control group) to 48% (Sr, control group). A statistical analysis of the censored data for the discussed measurements of serum elemental composition as regards patients fed parenterally was performed using censored-data statistical approach and discussed by us in the work [13], excluding Mn due to a very high censoring level (97%).

## Descriptive statistics for element concentrations in human serum samples: P, S, Cl, K, Ca, Cr, Fe, Cu, Zn, Se and Br

The aim of the presented study is to perform statistical analysis of the content of the remaining elements: P, S, Cl, K, Ca, Fe, Cu, and Zn, determined in the serum samples of the study and control groups. Since the analyses additionally include discussing the correlations between all elements, Cr, Se and Br, presented in [8], are also included in this paper.

Table 2 presents the value of the parameters of descriptive statistics for element concentration: P, S, Cl, K, Ca, Cr, Fe, Cu, Zn, Se and Br obtained, based on the raw results, in control group. The calculations were performed in the STATISTICA 13.3 [14].

Table 2 contains the following quantities: mean value, confidence interval, median, minimum, maximum, first and third quartiles, standard deviation and variation coefficient (defined as the ratio of the standard deviation to the mean value, in percent). Mean concentrations of elements are in the range from 0.054 mg/L (Cr) to 3190 mg/L (Cl). The dispersion of the element concentration distribution is described by the standard deviation and, in consequence, by variation coefficient in the range from about 6% to 70%. The smallest dispersion of concentration values is observed for macro elements: Cl (an

**Table 2. Descriptive statistics for element concentrations in serum of persons in control group.**

| Element | Descriptive statistics (control group) | | | | | | | | |
|---|---|---|---|---|---|---|---|---|---|
| | Mean value [mg/L] | Confidence interval (95%) [mg/L] | Median [mg/L] | Minimum [mg/L] | Maximum [mg/L] | First quartile [mg/L] | Third quartile [mg/L] | Standard deviation [mg/L] | Variation Coefficient [%] |
| P | 38.8 | 35.0 - 42.7 | 36.7 | 17.1 | 76.1 | 27.1 | 48.4 | 13.4 | 34.6 |
| S | 553 | 525 - 581 | 566 | 247 | 748 | 499 | 609 | 98.8 | 17.9 |
| Cl | 3190 | 3140 - 3240 | 3200 | 2700 | 3580 | 3080 | 3270 | 190 | 5.87 |
| K | 134 | 129 - 138 | 132 | 99.8 | 175 | 124 | 143 | 15.8 | 11.8 |
| Ca | 77.3 | 74.9 - 79.7 | 77.4 | 50.5 | 105 | 72.5 | 80.9 | 8.48 | 11.0 |
| Cr | 0.054 | 0.046 - 0.063 | 0.045 | 0.014 | 0.148 | 0.031 | 0.080 | 0.030 | 56.1 |
| Fe | 1.22 | 0.975 - 1.47 | 0.957 | 0.451 | 5.20 | 0.668 | 1.48 | 0.873 | 71.4 |
| Cu | 1.11 | 1.04 - 1.18 | 1.07 | 0.715 | 1.86 | 0.913 | 1.26 | 0.243 | 21.8 |
| Zn | 0.809 | 0.757 - 0.860 | 0.772 | 0.396 | 1.18 | 0.712 | 0.957 | 0.183 | 22.6 |
| Se | 0.056 | 0.051 - 0.060 | 0.055 | 0.023 | 0.121 | 0.047 | 0.062 | 0.015 | 27.3 |
| Br | 2.30 | 2.10 - 2.51 | 2.21 | 1.40 | 5.28 | 1.92 | 2.47 | 0.721 | 31.3 |

important electrolyte that regulates water-electrolyte and acid-base balance; is responsible for maintaining proper osmotic pressure and the functioning of the neuromuscular system), K (responsible for the proper functioning of the heart, nerve conduction and muscle contractions) and Ca (essential for the proper functioning of muscles, nerves, blood clotting and the formation of bones and teeth), for which the variation coefficient is in the range of 6–11%. The largest dispersion of concentration values is observed for Cr and Fe (variation coefficient 60–70%).

Table 3 presents parameter values of descriptive statistics for element concentration: P, S, Cl, K, Ca, Cr, Fe, Cu, Zn, Se, and Br obtained, based on the raw results, in studied group (measurement I and measurement II). The calculations were performed in the STATISTICA 13.3 [14].

Mean concentrations of elements are in the range from 0.050 mg/L (Cr) to 3230 mg/L (Cl). Dispersion of element concentration, described by variation coefficient, is in the range from about 7% to 220%. The smallest dispersion of concentration values is observed for Cl, K and Ca, for which the variation coefficient is in the range 7–18%. Larger dispersions of concentration values are observed for Cr, Se and Br (variation coefficient 60–70%). The largest dispersion, 140% (measurement I) and 220% (measurement II), is observed for Fe concentration. A comparison of the variation coefficient between the control and studied groups shows that the same groups of elements are characterized by the smallest and

**Table 3. Descriptive statistics for element concentrations in serum of patients in the studied group (measurement I and measurement II).**

| Element | Descriptive statistics (studied group, measurement I and measurement II) | | | | | | | | |
|---|---|---|---|---|---|---|---|---|---|
| | Mean value [mg/L] | Confidence interval (95%) [mg/L] | Median [mg/L] | Minimum [mg/L] | Maximum [mg/L] | First quartile [mg/L] | Third quartile [mg/L] | Standard deviation [mg/L] | Variation Coefficient [%] |
| measurement I | | | | | | | | | |
| P | 49.4 | 43.7 - 55.0 | 48.6 | 10.5 | 120 | 33.8 | 61.4 | 19.4 | 39.3 |
| S | 428 | 397 - 460 | 406 | 64.5 | 671 | 373 | 511 | 108 | 25.3 |
| Cl | 3230 | 3160 - 3300 | 3210 | 2720 | 3770 | 3080 | 3390 | 239 | 7.41 |
| K | 129 | 123 - 136 | 130 | 89.1 | 174 | 110 | 147 | 23.5 | 18.1 |
| Ca | 72.2 | 69.7 - 74.7 | 71.3 | 48.1 | 90.2 | 67.4 | 77.2 | 8.63 | 12.0 |
| Cr | 0.057 | 0.048 - 0.065 | 0.048 | 0.016 | 0.129 | 0.035 | 0.074 | 0.030 | 53.4 |
| Fe | 0.967 | 0.580 - 1.35 | 0.627 | 0.261 | 9.14 | 0.401 | 0.900 | 1.33 | 138 |
| Cu | 1.19 | 1.11 - 1.27 | 1.21 | 0.627 | 1.98 | 1.00 | 1.36 | 0.275 | 23.0 |
| Zn | 0.708 | 0.638 - 0.777 | 0.662 | 0.343 | 1.50 | 0.545 | 0.823 | 0.238 | 33.7 |
| Se | 0.045 | 0.040 - 0.051 | 0.044 | 0.011 | 0.106 | 0.031 | 0.056 | 0.019 | 42.2 |
| Br | 1.20 | 1.02 - 1.38 | 1.06 | 0.312 | 2.94 | 0.690 | 1.63 | 0.639 | 52.8 |
| measurement II | | | | | | | | | |
| P | 52.8 | 48.3 - 57.4 | 50.4 | 22.5 | 101 | 42.6 | 59.5 | 15.7 | 29.7 |
| S | 442 | 414 - 471 | 442 | 60.1 | 626 | 401 | 513 | 98.3 | 22.2 |
| Cl | 3220 | 3160 - 3300 | 3220 | 2440 | 3820 | 3050 | 3390 | 245 | 7.58 |
| K | 137 | 131 - 143 | 138 | 90.1 | 186 | 122 | 152 | 21.1 | 15.3 |
| Ca | 74.0 | 71.6 - 76.5 | 74.5 | 42.9 | 89.2 | 68.7 | 79.0 | 8.37 | 11.3 |
| Cr | 0.050 | 0.043 - 0.057 | 0.045 | 0.012 | 0.111 | 0.031 | 0.069 | 0.024 | 48.0 |
| Fe | 1.93 | 0.708 - 3.14 | 0.767 | 0.239 | 26.1 | 0.463 | 1.44 | 4.19 | 218 |
| Cu | 1.34 | 1.26 - 1.42 | 1.36 | 0.804 | 1.94 | 1.16 | 1.48 | 0.274 | 20.5 |
| Zn | 0.757 | 0.693 - 0.821 | 0.738 | 0.301 | 1.54 | 0.615 | 0.870 | 0.221 | 29.2 |
| Se | 0.052 | 0.046 - 0.058 | 0.053 | 0.018 | 0.110 | 0.037 | 0.062 | 0.021 | 40.4 |
| Br | 1.21 | 1.05 - 1.37 | 1.14 | 0.332 | 2.63 | 0.832 | 1.445 | 0.570 | 47.1 |

largest dispersion of their content. The dispersion is larger, however, for the studied group of people receiving parenteral nutrition, especially for S, K, Zn, Se, Br, and Fe.

The obtained results are presented graphically in Fig 1.

A detailed comparison of the content of elements between the control and studied groups (measurement I and measurement II) will be performed using statistical tests in the next sections of the presented paper.

### Distributions of element concentrations

Prior to verifying statistical hypotheses related to the element concentrations in control and studied groups, it was necessary to check the type of distribution that describes experimental values of the element content in the serum samples. Element distributions in the samples of human biological materials are usually described by lognormal distributions or, more generally, log-stable distributions [15,16]. These distributions are always asymmetric, with large standard deviations, suggesting a high population dispersion of elemental content. A characteristic feature of these distributions is the occurrence of extremely small or extremely large elemental content values (over three standard deviations, as tested for outliers), which is a natural observation. For the log-stable distributions, the best measures of location parameters are the median and quartiles because the presence of outliers does not change the values of these parameters [16].

Data presentation in the form of a histogram allows assessing theoretical distribution describing the experimental distribution. Fig 1S in S1 File (in Supporting information) presents a distribution of zinc content in the serum samples of patients from the studied group (measurement I). The red solid line represents lognormal distribution.

The $\chi^2$ goodness-of-fit test was used to verify the null hypothesis that experimental distribution of element concentration (here Zn concentration) is described by the lognormal distribution. The obtained test probability value $p = 0.4874$ ($p >$ significance level $\alpha = 0.05$) indicates that there is no evidence for rejecting the null hypothesis. Therefore, we conclude that the distribution of zinc content is described by the lognormal distribution. A feature of this distribution is that after the logarithmic transformation ($\ln x$) of the experimental values, a normal distribution is obtained. Fig 2S in S1 File

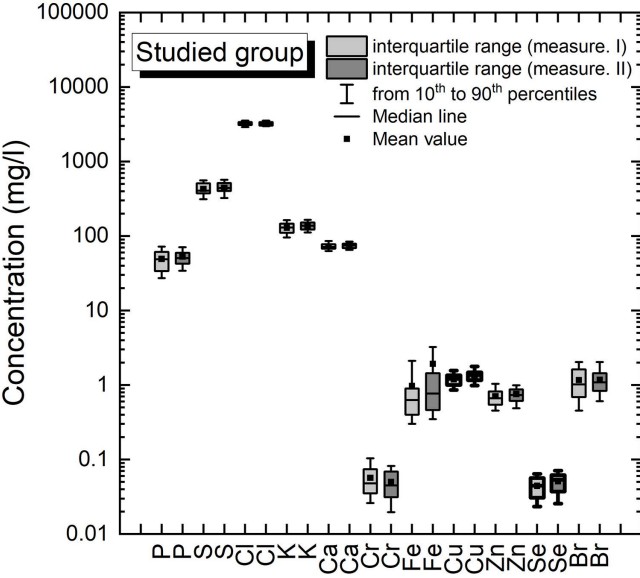

**Fig 1. Concentration of elements in serum samples of studied group for measurement I and measurement II. The box border is in bold for elements (Cu, Se) concentration different in measurement I and II (significance level $\alpha = 0.05$).**

(Supporting information) shows the same distribution of zinc content in serum samples of patients from the studied group (measurement I) obtained after logarithmic transformation. The red solid line represents normal distribution.

As in the previous case, the $\chi^2$ goodness-of-fit test was used to verify the null hypothesis that the logarithm of the experimental element concentration distribution is described by the normal distribution. The obtained test probability value $p = 0.8044$ ($p > \alpha = 0.05$) indicates that there is no evidence for rejecting the null hypothesis. Therefore, it can be concluded that the Zn content (after logarithmic transformation) is described by the normal distribution.

The $\chi^2$ goodness-of-fit test was performed for all element content distributions analyzed in the studied group (measurement I and measurement II) and in the control group. Table 1S in S1 File(Supporting information) presents the test probability p values with respect to the comparison of element content distributions with the lognormal distribution. The obtained p values confirm that there is no evidence for rejecting the null hypothesis. Therefore, element content distributions are described by lognormal distributions. In further statistical analyses, for the procedures that assume normal distribution, calculations were performed for the experimental values after the logarithmic transformation.

## Comparison of element concentration between women and men

Descriptive statistics and verifying the distribution type for the content of elements in the studied and control groups, presented in the previous section, were performed without taking into account patients' gender. Before verifying the hypotheses on comparing the content of elements between the studied and control groups, it had been analyzed whether the distributions of element contents depended on the gender of the patients within the groups.

Table 2S in S1 File presents a comparison of the mean value of element concentrations for women and men in the control group using a Student's t-test (t-test) for independent groups. In order to fulfill the assumption about the normality of random variable distribution (here element concentration) of the t-test, the calculations were performed on the logarithmically transformed data. The Table 2S in S1 File provides the mean value of the element content in control group of women and men, the p value for mean value (comparison of means), the number of valid cases in women group and in men group and the standard deviations for both groups. The last column contains the data with respect to the verification of variance equality in both groups. The p value for the variance is the test probability for variance comparing (assumption of the t-test for independent groups).

Based on the results obtained, it can be stated that the variances of element concentrations in the compared groups are the same (for all elements p for variance is higher than > 0.05). In the case of mean values, difference between women and men groups was observed only for sulfur concentration mean value (p value = 0.0449 is slightly smaller than 0.05).

Table 3S in S1 File shows a comparison of the mean value of element concentrations for women and men in the studied group (measurement I) using the t-test for independent groups. The results showed that only for sulfur the variances in the women and men were different (p = 0.0005). This means that in this case a nonparametric test should be used. For the rest of elements, no differences in the mean content of elements were observed between the compared groups.

Table 4S in S1 File shows a comparison of the mean value of element concentrations for women and men in the studied group (measurement II) using the t-test for independent groups. It was noted that for phosphorus and sulfur the variances among women and men were different (p = 0.0005). This means that in these cases a nonparametric test should be used. For other elements, no differences in the mean content of elements were observed between the compared groups. However, for Se it was observed that in measurement II the mean value of concentration is different for women and men. Considering that no difference was observed in measurement I and that selenium was supplemented, the difference in Se content in measurement II means different effectiveness of supplementation for the group of women and men.

In the case of elements for which the variances were different for women and men, i.e., S (measurement I), P (measurement II), S (measurement II), the nonparametric test of Mann-Whitney was applied to compare element concentration distributions. This test is the equivalent of the t-test for independent variables. The results of analysis are presented in

Table 5S in S1 File. The Table shows the sums of ranks, numbers of samples and p values. For each case, the p value is higher than α (p > α = 0.05). This means that there are no differences between the concentration distributions in the groups of women and men.

The observed differences in sulfur for the control group and selenium for the studied group (measurement II) concentrations based on gender suggest the potential need for gender-specific parenteral nutrition adjustments and baseline references.

## Discussion

### Comparison of the content of elements for the studied group before and after supplementation

Verification of the hypothesis related to the comparing the mean values of the element contents in the measurement I and measurement II of the studied group were performed using the t-test for dependent (paired) samples (STATISTICA). The verification concerns dependent samples because the same patients are taken into consideration before and after supplementation. T-test requires normal distribution, therefore the calculations were performed for the logarithmically transformed data. Additionally, in the first stage of comparisons, patient gender was not analyzed. In comparison, also a nonparametric sign test for the dependent variables was applied for the not transformed data.

The differences (p value < α), confirmed by both tests, were found for Cu and Se (Table 4). For P and K, the t-test showed statistical differences (with p values relatively close to 0.05) between element concentration in measurement I and measurement II, while the sign test, whose power is less than for t-test, did not confirm these differences.

**Table 4. P values for t-test for dependent samples in the group of patients receiving parenteral nutrition (studied group) without division by gender. In brackets, test p values for the nonparametric sign test for dependent variables are given.**

| Element/measurement | p value |
|---|---|
| ln P – measurement I | **0.0438** |
| ln P – measurement II | (0.1939) |
| ln S – measurement I | 0.2029 |
| ln S – measurement II | (0.6650) |
| ln Cl – measurement I | 0.9621 |
| ln Cl – measurement II | (0.6650) |
| ln K – measurement I | **0.0268** |
| ln K – measurement II | (0.1124) |
| ln Ca – measurement I | 0.3014 |
| ln Ca – measurement II | (0.8852) |
| ln Cr – measurement I | 0.1660 |
| ln Cr – measurement II | (0.8852) |
| ln Fe – measurement I | 0.0674 |
| ln Fe – measurement II | (0.1124) |
| ln Cu – measurement I | **0.0013** |
| ln Cu – measurement II | **(0.0141)** |
| ln Zn – measurement I | 0.0683 |
| ln Zn – measurement II | (0.1939) |
| ln Se – measurement I | **0.0060** |
| ln Se – measurement II | **(0.0141)** |
| ln Br – measurement I | 0.4200 |
| ln Br – measurement II | (0.8852) |

Next, the gender of the patients was also taken into account in the comparisons. Tables 5 and 6 present the results for men and women, respectively. In the case of men, no differences, confirmed by both tests, were observed in element contents between measurement I and II (Table 5). The p value close to 0.05 (p = 0.0503, t-test) was observed for Cu (Table 5).

In the case of women, similarly to the statistical analysis without taking the patient's gender into consideration, the differences (p value < α) confirmed by both tests were found for Cu and Se (Table 6). This was not observed, however, for the group of men. Additionally, for Fe and Zn concentrations, the differences between measurement I and II were confirmed with one of the applied tests.

In summary, it can be stated that large differences in the content of element concentrations between measurement I and II were observed in the case of Se (increased by 23% after parenteral supplementation, from 0.047 mg/l to 0.058 mg/l) and Cu (increased by 16% after parenteral supplementation, from 1.16 mg/l to 1.34 mg/l) for the group of women. The differences in the group of women result in differences in the content of these elements (Cu and Se) in the analysis of all samples (without considering the gender of the patients).

Increased Cu and Se concentrations after parenteral nutrition and the relative stability of Zn and Fe can be interpreted in the context of their specific metabolic roles, transport proteins and supplementation practices used during parenteral nutrition. The increase in Cu and Se after parenteral nutrition can result directly from their supplementation and their rapid incorporation into plasma proteins. Copper responds through increased ceruloplasmin synthesis, which rises quickly once

**Table 5. P values for t-test for dependent samples in the group of patients receiving parenteral nutrition (studied group) performed for men samples. Test p values for the nonparametric sign test for dependent variables are given in brackets.**

| Element/measurement | p value |
|---|---|
| ln P – measurement I | 0.1897 |
| ln P – measurement II | (0.7003) |
| ln S – measurement I | 0.7071 |
| ln S – measurement II | (0.7003) |
| ln Cl – measurement I | 0.6010 |
| ln Cl – measurement II | (0.2482) |
| ln K – measurement I | 0.1829 |
| ln K – measurement II | (0.7003) |
| ln Ca – measurement I | 0.4766 |
| ln Ca – measurement II | (1.000) |
| ln Cr – measurement I | 0.2034 |
| ln Cr – measurement II | (1.000) |
| ln Fe – measurement I | 0.2783 |
| ln Fe – measurement II | (1.000) |
| ln Cu – measurement I | 0.0503 |
| ln Cu – measurement II | (0.2482) |
| ln Zn – measurement I | 0.6115 |
| ln Zn – measurement II | (1.000) |
| ln Se – measurement I | 0.1122 |
| ln Se – measurement II | (0.4414) |
| ln Br – measurement I | 0.5031 |
| ln Br – measurement II | (0.7003) |

metabolic status improves during parenteral nutrition. Selenium concentration reflects efficient incorporation of selenite into selenoproteins such as glutathione peroxidase, a process enhanced by controlled parenteral nutrition delivery.

In contrast, Zn and Fe remain stable because their serum levels are tightly regulated. Zinc is strongly controlled by albumin binding and redistribution to tissues, so supplementation does not markedly change serum Zn. Iron is not routinely supplemented in parenteral nutrition, and inflammatory regulation through hepcidin further stabilizes or suppresses serum Fe release.

## Comparison of the content of elements for the studied group with the control group

A comparison of element concentrations obtained for the studied group in measurement I and measurement II with the values for control group are presented in the Table 6S in S1 File. A statistical analysis was performed for the logarithmically transformed data (regardless of gender) using the t-test for independent samples.

The null hypothesis assumes that there are no statistically significant differences between the mean value of the element content in the studied group (measurement I/measurement II) and in the control group. The alternative hypothesis assumes that the mean contents are different. Table 6S in S1 File presents p values for the performed hypotheses verification. In the case of S, K, Se and Br (comparisons of measurement I and control groups and measurement II and control groups), Fe (comparison of measurement II and control groups), verifications needed an additional application of the nonparametric Mann-Whitney test due to the fact that variances in compared groups were different (p value for variance less than 0.05). Equality of variances is an assumption of the t-test.

**Table 6. P values for the t-test for dependent samples in the group of patients receiving parenteral nutrition (studied group) performed for women samples. Test p values for the nonparametric sign test for dependent variables are given in brackets.**

| Element/measurement | p value |
|---|---|
| ln P – measurement I | 0.1278 |
| ln P – measurement II | (0.1904) |
| ln S – measurement I | 0.1545 |
| ln S – measurement II | (0.1904) |
| ln Cl – measurement I | 0.5009 |
| ln Cl – measurement II | (0.6625) |
| ln K – measurement I | 0.0507 |
| ln K – measurement II | (0.0809) |
| ln Ca – measurement I | 0.3906 |
| ln Ca – measurement II | (1.000) |
| ln Cr – measurement I | 0.5569 |
| ln Cr – measurement II | (1.000) |
| ln Fe – measurement I | 0.0839 |
| ln Fe – measurement II | **(0.0088)** |
| ln Cu – measurement I | **0.0112** |
| ln Cu – measurement II | **(0.0291)** |
| ln Zn – measurement I | **0.0409** |
| ln Zn – measurement II | (0.0809) |
| ln Se – measurement I | **0.0271** |
| ln Se – measurement II | **(0.0088)** |
| ln Br – measurement I | 0.6677 |
| ln Br – measurement II | (1.000) |

Table 7S in S1 File presents the p values for the Mann-Whitney test.

The tests show that the average content of elements between the studied (measurement I) and the control group is different for the following elements: P, S, Ca, Fe, Zn, Se, and Br. For the studied group (measurement II) the differences are observed for: P, S, Cu, and Br. Based on the results of element contents for the studied and control groups (Tables 2 and 3) and the results of statistical tests, it can be concluded that the use of parenteral nutrition in the studied group caused a statistically significant change in the content of Ca, Fe, Cu, Zn, and Se elements.

Graphical presentation of the obtained results for Cu is presented in the Figs 2 (logarithmically transformed data) and 3. Figs 2 and 3 illustrate raw data, concentration mean values and standard deviations for each group. The p value (p = 0.0013) was related to the comparison of Cu concentration in the studied group for measurement I and II. In addition, Figs 2 and 3 present the reference value range of Cu concentration in the human serum, used in the routine analysis of serum samples of patient of the Holycross Cancer Center [10].

A comparison of the element content for studied group with the control group was also performed separately for women and men with the application of the same statistical procedures. Tables 8S in S1 File (p values for t-test) and 9S (p values for nonparametric Mann-Whitney test) present the results for women.

The tests show that the average content of elements between the studied group of women (measurement I) and the control group is different for the following elements: P, S, Ca, Zn, Se, and Br. For measurement II (group of women), the differences are observed for the following elements: P, S, Cu, and Br. It can be concluded that the use of parenteral nutrition in the studied group of women caused a statistically significant change in the content of Ca, Cu, Zn, and Se.

Tables 10S in S1 File (p values for t-test) and 11S (p values for the nonparametric Mann-Whitney test) present the results for the group of men. The tests show that the average content of elements between the studied men (measurement I) and the control group is different for the following elements: S, Ca, Cr, Fe, Cu, Se and Br. In the case of

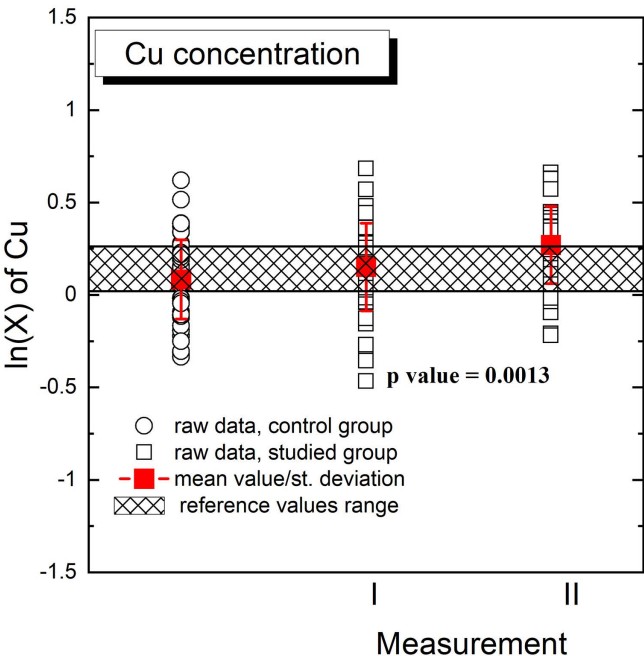

**Fig 2. Graphical presentation of comparison of Cu concentrations in the control and studied (I and II measurement) groups.** Figure illustrates raw data, concentration mean values and standard deviations for each group and the reference value range of Cu concentration in the human serum. The p value (p = 0.0013) was related to the comparison of Cu concentration in the studied group for measurement I and measurement II. The data are presented after the logarithmic transformation.

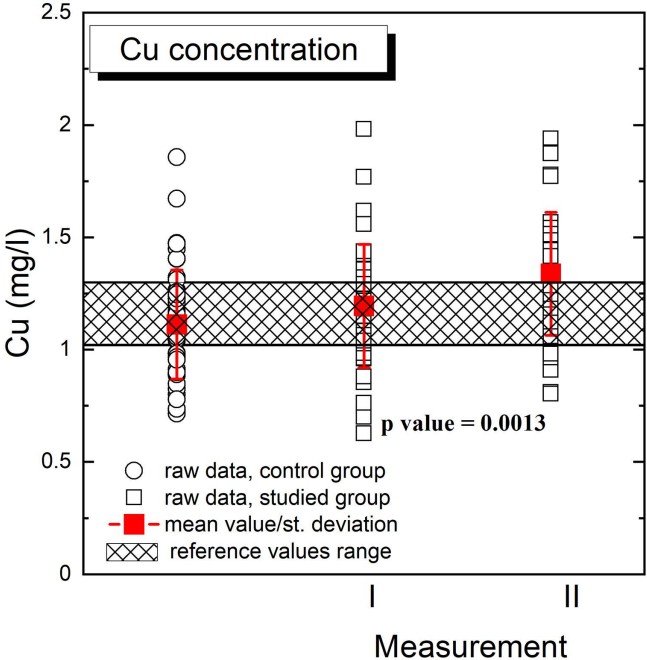

**Fig 3. Graphical presentation of comparison of Cu concentrations in the control and studied (measurement I and measurement II) groups.** Figure shows raw data, concentration mean values and standard deviations for each group and the reference value range of Cu concentration in the human serum. The p value (p = 0.0013) was related to the comparison of Cu concentration in the studied group for measurement I and measurement II.

measurement II, the different values can be noticed for: P, S, Cu, Se, and Br. Similarly, it can be concluded that the use of parenteral nutrition in the studied group of men caused a statistically significant change in the content of P, Ca, Cr, Fe, and Cu (i.e., a change in the p value).

## Analysis of correlations between elements

In the presented work, correlations between elements in the parenterally fed group and the control group were determined using the Pearson correlation coefficient. This coefficient is used to check whether there is a linear relationship between two quantitative variables. The use of this correlation requires that the variables that are analyzed have a normal distribution. In order to fulfil this assumption, the calculations were performed for the values of element content after the logarithmic transformation, which ensured the normality of the distributions of element concentration. Table 12S in S1 File presents the correlation coefficients between elements in the control group. The Pearson's correlation coefficients statistically different than zero are marked in bold.

A statistically significant and positive value of the correlation coefficient was obtained for the following pairs of elements: P and Ca, P and Cu, S and Ca, S and Fe, S and Fe, S and Se, Cl and K, Cl and Ca, K and Zn, K and Br, Ca and Cu, Ca and Zn, Ca and Se, Fe and Zn, Zn and Se, Zn and Br, Se and Br. The value of this parameter is in the range from 0.2806 to 0.7204. A positive value of the correlation coefficient means that an increase in the content of one element is accompanied by an increase in the content of the other element. Negative values of the correlation coefficient were observed less frequently. Such statistically significant values were obtained for the following pairs of elements: S and Cr (correlation coefficient −0.5148), Ca and Cr (−0.4049). A negative value of the correlation coefficient means that an increase in the content of one element is accompanied by a decrease in the content of another element.

Table 13S in S1 File presents the correlation coefficients between elements in studied group obtained for measurement I. A statistically significant and positive value of the correlation coefficient was obtained for the following pairs of elements: P and Ca, S and Ca, Cl and Br, K and Ca, Ca and Cu, Ca and Se, Fe and Br, Cu and Zn, Cu and Se, Zn and Se. The value of this parameter is in the range from 0.2890 to 0.6520. Negative values of the correlation coefficient were observed less frequently. Such statistically significant values were obtained for the following pairs of elements: P and Cr (correlation coefficient −0.3001), Ca and Cr (−0.4596).

Table 14S in S1 File presents the correlation coefficients between elements in studied group obtained for measurement II. A statistically significant and positive value of the correlation coefficient was obtained for the following pairs of elements: P and Cu, S and Ca, S and Zn, Cl and K, Cl and Ca, Cl and Br, K and Ca, Cu and P, Cu and Zn, Cu and Se, Zn and Se. The value of this parameter is in the range from 0.2871 to 0.6173. Negative values of the correlation coefficient were observed less frequently. Such statistically significant values were obtained for the following pairs of elements: S and Cr (correlation coefficient −0.3342), Ca and Cr (−0.2881).

Summarizing the results for the three analyzed groups, i.e., the control group, the studied group for measurement I and the studied group for measurement II, it was observed that in all groups the positive correlations were obtained between S and Ca. The correlation coefficient was the highest for the control group (about 0.7) and comparable in the studied groups (about 0.4)) and for Zn and Se (in all groups the correlation coefficient was about 0.4–0.5). In all of the above-mentioned groups, there was a negative correlation between Ca and Cr (correlation coefficient from about −0.5 to −0.3). The positive correlation between sulfur (S) and calcium (Ca) concentration observed in all groups may indicate interdependence in metabolic changes, especially in the context of structural components of tissues and enzymatic processes but parenteral nutrition formulation may also contribute in this correlation. Sulfur is present, among others, in sulfur amino acids (cysteine, methionine), which affect calcium homeostasis, among others, by binding and storing $Ca^{2+}$ ions in plasma proteins. This may also suggest joint regulation of transport or dependence on the nutritional status of patients [17]. Parenteral nutrition mixtures provide amino acids and electrolytes in fixed proportions, which may reinforce this coordinated pattern. The effects mentioned above explain the persistence of a positive Ca–S relationship across all measurements and in the control group. Similarly, the positive correlation between zinc (Zn) and selenium (Se) is consistent with the literature, which indicates the synergistic effect of both elements in protection against oxidative stress. Both Zn and Se are essential for the proper functioning of the immune system and play a key role in the action of antioxidant enzymes [18]. In turn, the negative correlation between calcium (Ca) and chromium (Cr) may suggest potential competition in ion transport mechanisms or an antagonistic effect in the context of their metabolic role. Chromium participates in the regulation of carbohydrate metabolism, and its transport may occur independently of, or against, calcium homeostasis [19].

Finally, the correlations between element concentrations measured for the studied group in measurement I and measurement II were calculated. The results are presented in Table 15S in S1 File (Supporting information). For most elements, a positive, statistically significant correlation was demonstrated between the content of a given element determined in the first and the second measurement. The highest value of the correlation coefficient is observed for Br. For some elements, i.e., Ca, Cr, and Fe, the value of the content in the second measurement is not dependent on the content in the first measurement.

### Prediction of trace element concentrations following seven days of parenteral nutrition therapy

The results of the elemental analysis of serum samples of the studied group, conducted using the TXRF technique, combined with information about anthropometric, biochemical and immunological parameters can give a comprehensive view of the patient's condition. The presented study employs machine learning-based regression approaches to quantitatively model the relationship between clinical variables and the concentration of the selected trace elements in patients' blood serum. Regression algorithms enable predicting a continuous variable (dependent variable) based on a set of explanatory features. In predicting element concentrations in measurement II (output data), the following characteristics were taken

into account as input data: patient's age, gender, weight and height, weight loss within 3–6 months [%], body weight 6 months ago [kg], hip circumference [cm], waist to hip ratio (WHR) [cm], skin fold thickness over the triceps muscle of the non-dominant arm (TSF) [cm], circumference of the arm measured halfway between the coracoid and the ulna (MAC) [cm], arm muscle circumference (OMR) [cm], CRP [mg/L], albumin [g/dl], total protein [g/dl], level of leukocytes (WBC) [thousands per liter of blood], level of lymphocytes [%], total lymphocyte count (CLL), daily nitrogen excretion [g/24h], daily urea excretion grams [g], Nutritional Risk Index (NRI, a numerical score calculated to assess a patient's risk of malnutrition; is based on serum albumin levels and the ratio of current body weight to ideal body weight), BMI, element's concentration in measurement I: Na, Mg, P, Cl, K, Ca and Fe concentrations in blood.

The dataset was divided into two parts: a training set and a test set. This division facilitates optimizing model parameters on the training data while independently validating model performance on the unseen test data. This approach mitigates the risk of overfitting, a scenario in which a model performs well on training data but generalizes poorly to new data. Prior to model training, standardization was applied due to the sensitivity of regression algorithms to variable scaling.

In the predictions the following regression algorithms were applied: Lasso (L1 Regularization), Ridge Regression, Elastic Net, Random Forest Regression and XGBoost (Extreme Gradient Boosting) [20–26]. Lasso regression (Least Absolute Shrinkage and Selection Operator) introduces L1 regularization to the loss function, encouraging sparsity in model coefficients. This results in the automatic exclusion of irrelevant features, enhancing model interpretability and generalization, particularly useful in clinical datasets with potentially redundant predictors. Ridge regression augments linear regression with an L2 penalty, which reduces coefficient magnitudes to control model complexity and combat overfitting, while retaining all predictors in the model. Elastic Net combines the penalties of the Lasso and Ridge regression, offering a balanced approach suitable for datasets with high-dimensional, correlated predictors. It enables both feature selection and grouping effects, making it particularly appropriate for biomedical datasets with numerous interdependent variables. Random Forest is an ensemble method constructing multiple decision trees and aggregating their outputs to generate robust, stable predictions. For regression tasks, it predicts continuous values by averaging results from all trees. This method benefits from the reduced variance and improved accuracy, especially in handling nonlinearities and interactions among variables. XGBoost is a gradient boosting algorithm which sequentially builds trees to correct the errors of the previous ones. It integrates regularization, handles missing values, and supports outlier detection and categorical variables. Due to its efficiency and predictive performance, it is widely applied in the structured biomedical datasets.

To assess model performance, the Mean Squared Error (MSE) and Coefficient of Determination ($R^2$) metrics were employed. The MSE parameter measures an average squared difference between the predicted and actual values. Lower MSE value indicates a better model fit. $R^2$ quantifies the proportion of variance in the dependent variable explained by the model. $R^2$ values close to 1 signify excellent model performance; values below 0 indicate poor fit, worse than random guessing.

The prediction of trace element concentrations following seven days of parenteral nutrition therapy was performed for all the studied elements. The best results (low MSE and the highest $R^2$ values) were obtained for Br. Table 7 summarizes

**Table 7. MSE and $R^2$ values for the applied regression algorithms of predicting Br concentration in measurement II.**

| Regression algorithms | Br | |
|---|---|---|
| | MSE | $R^2$ |
| Lasso (L1 Regularization) | 0.171 | 0.47 |
| Ridge Regression | 0.303 | 0.06 |
| **Elastic Net** | **0.158** | **0.51** |
| Random Forest Regression | 0.178 | 0.45 |
| XGBoost (Extreme Gradient Boosting) | 0.263 | 0.18 |

the results for this element and applied regression algorithms. As can be seen from the table, the best prediction of Br concentration in measurement II was achieved for the Elastic Net algorithm.

Fig 4 presents the results of predicting Br concentrations following seven days of parenteral nutrition therapy. In predicting Br content in measurement II, the following variables measured in measurement I are considered as the most significant: Br concentration (importance 50%), total lymphocyte count (CLL) (12.7%), level of leukocytes (WBC) (10.2%), K concentration (4.54%), Cl concentration (2.62%), and total protein (2.05%). Importance of other variables is below 2%.

## Conclusions

This paper discusses the study of element concentrations in blood serum of patients receiving parenteral nutrition using the total reflection X-ray fluorescence analysis technique. For comparative purposes, also the control group was taken into consideration. For both groups, the contents of the following macro- and microelements were presented comprehensively: P, S, Cl, K, Ca, Cr, Fe, Cu, Zn, Se, and Br. The conducted statistical analysis focused on determining descriptive statistics and checking the distribution type of the elements' content. It was observed that the dispersion of the values as regards the element content is smaller for macro elements and larger for trace elements, especially for Cr and Fe. Additionally, the dispersion of element concentration is larger for the studied group of patients receiving parenteral nutrition. A characteristic feature of the element concentration distributions is the occurrence of extremely small or extremely large elemental content values, which is a natural observation. For these distributions, the best measures of location parameters are the median and quartiles.

The observed differences in sulfur for the control group and selenium for the studied group (measurement II) concentrations based on gender suggest the potential need for gender-specific parenteral nutrition adjustments and baseline references.

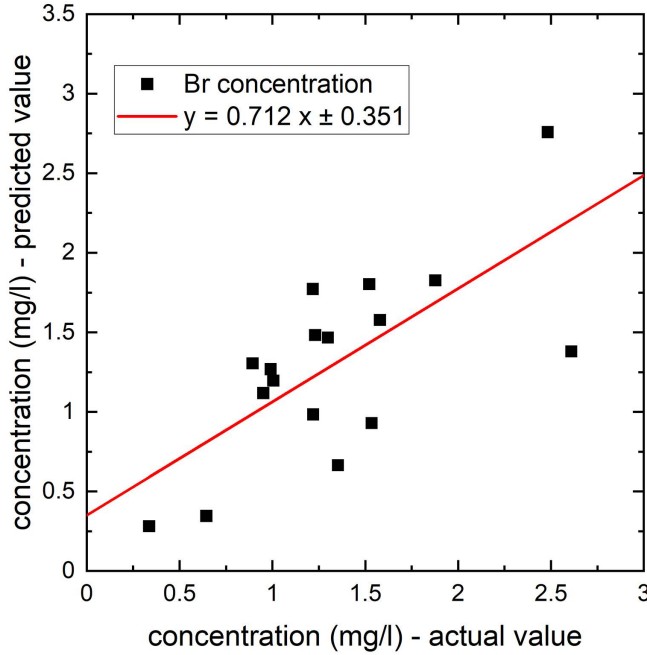

**Fig 4. The results of predicting Br concentrations following seven days of parenteral nutrition therapy.**

The objective was also to verify statistical hypotheses concerning the comparison of the elements' content in the study and control groups and to compare measurement results I and II in the study group. It was found out that for some elements the differences between the studied (measurement I) and control groups can be observed, which can be eliminated by supplementation of the elements during parenteral nutrition. The statistical significance differences in the content of element concentrations between measurement I and II were observed for the group of women in the case of Se (increased by 23% after parenteral supplementation) and Cu (increased by 16% after parenteral supplementation). The differences in the group of women result in differences in the content of these elements (Cu and Se) in the analysis of all samples (without considering the gender of the patients). Increased Cu and Se concentrations after parenteral nutrition can be interpreted in the context of their specific metabolic roles, transport proteins and supplementation practices used during parenteral nutrition.

In all groups, positive correlations were obtained between S and Ca (coefficient on the level 0.4 for the studied group and 0.7 for the control group. Such values were additionally obtained for Zn and Se (in all groups the correlation was about 0.4–0.5). In all groups there was a negative correlation between Ca and Cr (correlation coefficient ranged from −0.5 to −0.3).

In this studies, the prediction of trace element concentrations following seven days of parenteral nutrition therapy was also studied on the basis of knowledge of various anthropometric, biochemical, and immunological parameters describing the patient's condition.

In conclusion, it can be stated that the use of comprehensive information obtained from the simultaneous analysis of the element contents in serum using the TXRF technique and from statistical analysis, enables assessing the health status of the patients receiving parenteral nutrition and planning more personalized parenteral nutrition protocols. In the future, it is planned to increase the number of patients in the study and control groups and to improve the gender balance.

## Supporting information

**S1 File. Supplementary materials for statistical analysis.**
(PDF)

## Author contributions

**Conceptualization:** Monika Pierzak-Stępień, Stanisław Głuszek.

**Data curation:** Monika Pierzak-Stępień.

**Formal analysis:** Monika Pierzak-Stępień, Aldona Kubala-Kukuś, Dariusz Pasieka, Natalia Wojtaś, Stanisław Głuszek.

**Funding acquisition:** Monika Pierzak-Stępień, Dariusz Banaś, Stanisław Głuszek.

**Investigation:** Monika Pierzak-Stępień, Ilona Stabrawa, Jolanta Wudarczyk-Moćko, Karol Szary.

**Methodology:** Monika Pierzak-Stępień.

**Project administration:** Monika Pierzak-Stępień, Dariusz Banaś.

**Resources:** Monika Pierzak-Stępień, Aldona Kubala-Kukuś, Dariusz Banaś, Stanisław Głuszek.

**Software:** Dariusz Pasieka.

**Supervision:** Monika Pierzak-Stępień.

**Validation:** Monika Pierzak-Stępień, Aldona Kubala-Kukuś, Jolanta Wudarczyk-Moćko.

**Visualization:** Monika Pierzak-Stępień, Aldona Kubala-Kukuś, Milena Piotrowska.

**Writing – original draft:** Aldona Kubala-Kukuś, Dariusz Banaś, Monika Biernacka, Milena Piotrowska, Dariusz Pasieka, Andrzej Dąbrowski.

**Writing – review & editing:** Monika Pierzak-Stępień, Aldona Kubala-Kukuś, Dariusz Banaś, Ilona Stabrawa, Monika Biernacka, Milena Piotrowska, Dariusz Pasieka.

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
