## [Decision Letter · Decision Letter 0]

26 Oct 2025

Dear Dr. Kubala-Kukuś,

Thank you for submitting your manuscript to PLOS ONE. After careful consideration, we feel that it has merit but does not fully meet PLOS ONE’s publication criteria as it currently stands. Therefore, we invite you to submit a revised version of the manuscript that addresses the points raised during the review process.

We look forward to receiving your revised manuscript.

Kind regards,

Satish Rojekar, Ph.D.

Academic Editor

PLOS ONE

Journal Requirements:

“The functioning of the facility is supported by Polish Ministry of Education and Science (project 28/489259/SPUB/SP/2021).

This work was co-financed by the Minister of Science (Poland) under the "Regional Excellence Initiative" program (project no.: RID/SP/0015/2024/01).”

“The functioning of the facility is supported by Polish Ministry of Education and Science (project 28/489259/SPUB/SP/2021).

This work was co-financed by the Minister of Science (Poland) under the "Regional Excellence Initiative" program (project no.: RID/SP/0015/2024/01).”

“The functioning of the facility is supported by Polish Ministry of Education and Science (project 28/489259/SPUB/SP/2021).

This work was co-financed by the Minister of Science (Poland) under the "Regional Excellence Initiative" program (project no.: RID/SP/0015/2024/01).”

Reviewers' comments:

Reviewer's Responses to Questions

**Comments to the Author**

1. Is the manuscript technically sound, and do the data support the conclusions?

Reviewer #1: Yes

Reviewer #2: Yes

Reviewer #3: Yes

Reviewer #4: Partly

2. Has the statistical analysis been performed appropriately and rigorously?

Reviewer #1: Yes

Reviewer #2: Yes

Reviewer #3: Yes

Reviewer #4: Yes

3. Have the authors made all data underlying the findings in their manuscript fully available?

Reviewer #1: Yes

Reviewer #2: Yes

Reviewer #3: Yes

Reviewer #4: Yes

4. Is the manuscript presented in an intelligible fashion and written in standard English?

Reviewer #1: Yes

Reviewer #2: Yes

Reviewer #3: Yes

Reviewer #4: Yes

Reviewer #1: Great work - detailed study was performed and results were analyzed. A few points to consider:

1. It would be valuable to briefly mention potential directions for future research. For example, could the study be expanded to other patient populations, or might it be used to refine more personalized parenteral nutrition protocols?

2. A brief mention of any limitations or potential sources of bias would be beneficial. Discussing aspects such as sample size, study duration, or patient demographics could help contextualize the results and provide a balanced perspective.

Reviewer #2: This article presents a comprehensive study of elemental concentrations in the blood serum of patients undergoing parenteral nutrition, using total reflection X-ray fluorescence (TXRF) analysis. The study analyzes major and trace elements, supplementation effects, gender differences, and applies machine learning. Comments below aim to improve clarity, transparency and clinical relevance.

1. Abstract should more clearly state the key findings and clinical significance rather than extensive methodological details.

2. In the introduction, background on parenteral nutrition is comprehensive but it’s too lengthy. It could be revised to highlight the key research gap.

3. The study group is relatively small (n=48), and gender/age distribution between study and control groups is not perfectly balanced. This should be discussed as a limitation.

4. The method is well described, but reproducibility and quality control data should be more clearly summarized. For instance, detection limits and uncertainty levels are scattered in the text. These should be presented in a concise table.

5. The clinical implications of the findings are underdeveloped. For example, how might the observed Se and Cu differences affect nutritional protocols or patient monitoring? Are these differences clinically significant, or only statistically significant?

6. Results are heavily tabulated. Adding graphical visualizations (boxplots, forest plots, heatmaps) would improve clarity.

7. Ensure consistent use of abbreviations (e.g., CRP, NRS, BMI).

Reviewer #3: Thank you for the opportunity to review this manuscript. Overall, the study is technically sound, methodologically rigorous, and contributes valuable data on trace-element monitoring using TXRF in patients receiving parenteral nutrition. The statistical analysis is appropriate, the data appear robust, and the conclusions are supported by the results. I recommend Minor Revision, primarily to improve clarity in methodological details, data presentation, and biological interpretation.

A detailed review with specific comments and suggestions has been provided in the attached document.

Reviewer #4: 1. Refine the Abstract's Focus: Streamline the abstract to highlight the main hypothesis and two or three most impactful findings. For example, focus on the differences between baseline (Meas. I) and post-PN (Meas. II) for the supplemented elements (Cr, Se) and for any critical elements found to be deficient/excessive. Do not include specific correlation coefficients (0.4-0.7) or an exhaustive list of all differing elements (P, S, Ca, Fe...).

2. Clearly State the Direction of Change: When mentioning differences in the abstract, clarify the direction. Instead of, "Comparing element concentrations for measurement I and II indicates the differences in Cu and Se," write, "Parenteral nutrition resulted in a significant increase in supplemented Se and a decrease in Cu concentrations between measurement I and II."

3. Strengthen Control Group Justification: In the Introduction or Methods, briefly state why the cholecystectomy patients were deemed an appropriate control. This bolsters the argument that their elemental levels represent a healthy baseline, despite being hospitalized for surgery.

4. For the Methods and Results Sections

Enhance Data Visualization: The presentation of descriptive statistics (Tables 1 and 2) is detailed, but the results section would be far more accessible with clear visualization, such as box-and-whisker plots or bar charts. This is essential for illustrating the distributions, outliers (especially for Fe, as indicated by the high standard deviation), and group differences (Control vs. Studied I vs. Studied II).

5. Address High Variability (Fe): The extremely high variation coefficients for Iron (Fe) in the studied group (138% and 218%) are concerning. This suggests the presence of significant outliers or heterogeneous patient conditions (e.g., patients with severe anemia or recent blood transfusions). The authors must address the likely source of this high variability and justify whether the outliers were included or excluded and why.

6 Explicitly Link Elemental Changes to PN Content: Given the focus on PN, the results discussion should directly compare the measured changes to the composition of the administered PN mixture. For instance, if Se increased, confirm that the supplementation amount was appropriate for the observed change. If Cu changed, discuss if this was an intended or unintended effect of the total nutritional mixture.

7. Clarity on Gender Comparison: The text mentions a difference in Sulfur (S) for the control group and Selenium (Se) for the studied group (measurement II) based on gender. This finding should be integrated more fully into the discussion, as it suggests the potential need for gender-specific PN adjustments or baseline references.

**Do you want your identity to be public for this peer review?** For information about this choice, including consent withdrawal, please see our Privacy Policy

Reviewer #1: No

Reviewer #2: No

Reviewer #3: **Yes:** Rashi Porwal

Reviewer #4: No

---

## [Author Response · Author response to Decision Letter 1]

24 Nov 2025

Kielce, 17.11.2025

Satish Rojekar, Ph.D.

Academic Editor

PLOS ONE

Journal: PLOS ONE

Manuscript Number: PONE-D-25-39141

Title: Study of element concentrations in blood serum of patients receiving parenteral nutrition using total reflection X-ray fluorescence analysis

Authors: Monika Pierzak-Stępień, Aldona Kubala-Kukuś, Dariusz Banaś, Ilona Stabrawa, Jolanta Wudarczyk-Moćko, Karol Szary, Monika Biernacka, Milena Piotrowska, Dariusz Pasieka, Natalia Wojtaś, Andrzej Dąbrowski, Stanisław Głuszek

Authors would like to thank the Editor and Reviewers for their accurate comments and suggestions which taking into account in the manuscript have essentially improved the quality of the paper.

Note: The manuscript lines quoted in response to the reviewers refer to the revised manuscript without track changes.

Journal Requirements:

Question 1. Please ensure that your manuscript meets PLOS ONE's style requirements, including those for file naming. The PLOS ONE style templates can be found at

Authors' response: Thank to the editor for this note. The manuscript was checked in context of PLOS ONE’s style reguirements.

Question 2. Thank you for stating the following financial disclosure:

“The functioning of the facility is supported by Polish Ministry of Education and Science (project 28/489259/SPUB/SP/2021).

This work was co-financed by the Minister of Science (Poland) under the "Regional Excellence Initiative" program (project no.: RID/SP/0015/2024/01).”

Authors' response: Thank to the editor for this note. The funders had no role in the study. We state: "The funders had no role in study design, data collection and analysis, decision to publish, or preparation of the manuscript." This information was included in the Cover letter.

Question 3. Thank you for stating the following in the Acknowledgments Section of your manuscript:

“The functioning of the facility is supported by Polish Ministry of Education and Science (project 28/489259/SPUB/SP/2021).

This work was co-financed by the Minister of Science (Poland) under the "Regional Excellence Initiative" program (project no.: RID/SP/0015/2024/01).”

“The functioning of the facility is supported by Polish Ministry of Education and Science (project 28/489259/SPUB/SP/2021).

This work was co-financed by the Minister of Science (Poland) under the "Regional Excellence Initiative" program (project no.: RID/SP/0015/2024/01).”

Authors' response: Thank to the editor for this note. We removed the funding information from Acknowledgments section of our manuscript. In the Cover letter we specified the information we wanted to include in the Funding Statement.

Question 4. If the reviewer comments include a recommendation to cite specific previously published works, please review and evaluate these publications to determine whether they are relevant and should be cited. There is no requirement to cite these works unless the editor has indicated otherwise.

Authors' response: Thank to the editor for this note. The reviewer comments did not include a recommendation to cite specific previously published works.

Question 5. To ensure your figures meet our technical requirements, please review our figure guidelines: https://journals.plos.org/plosone/s/figures

Authors' response: Thank to the editor for this note. The figures were checked and corrected.

Reviewers' comments to the Author:

Reviewer #1

Question 6. It would be valuable to briefly mention potential directions for future research. For example, could the study be expanded to other patient populations, or might it be used to refine more personalized parenteral nutrition protocols?

Authors' response: Thank to the reviewer for this note. Information was added in the Conclusions.

In fact, the multi-element TXRF profiling of human serum samples is used by use in routine, daily, analysis of serum of Holycross Cancer Center patients (Kielce, Poland). We typically analyze the serum of individuals with Hodgkin's lymphoma for copper content and the serum of patients treated with selenium preparations.

Question 7. A brief mention of any limitations or potential sources of bias would be beneficial. Discussing aspects such as sample size, study duration, or patient demographics could help contextualize the results and provide a balanced perspective.

Authors' response: Thank to the reviewer for this note. In the conclusions the following sentence was added: ,,In the future, it is planned to increase the number of patients in the study and control groups and to improve the gender balance.” (line 756).

Reviewer #2

Question 8. Abstract should more clearly state the key findings and clinical significance rather than extensive methodological details.

Authors' response: Thank to the reviewer for this note. Abstract was corrected.

Question 9. In the introduction, background on parenteral nutrition is comprehensive but it’s too lengthy. It could be revised to highlight the key research gap.

Authors' response: Thank to the reviewer for this note. Introduction was corrected.

Question 10. The study group is relatively small (n=48), and gender/age distribution between study and control groups is not perfectly balanced. This should be discussed as a limitation.

Authors' response: Thank to the reviewer for this note. In the conclusions the following sentence was added: ,,In the future, it is planned to increase the number of patients in the study and control groups and to improve the gender balance.” (line 756).

Question 11. The method is well described, but reproducibility and quality control data should be more clearly summarized. For instance, detection limits and uncertainty levels are scattered in the text. These should be presented in a concise table.

Authors' response: Thank to the reviewer for this note. We prepared table with discussed information (Table 1).

Question 12. The clinical implications of the findings are underdeveloped. For example, how might the observed Se and Cu differences affect nutritional protocols or patient monitoring? Are these differences clinically significant, or only statistically significant?

Authors' response: Thank to the reviewer for this note. In fact, the multi-element TXRF profiling of human serum samples is used by use in routine, daily, analysis of serum of Holycross Cancer Center patients (Kielce, Poland). We typically analyze the serum of individuals with Hodgkin's lymphoma for copper content and the serum of patients treated with selenium preparations. Therefore, implementation of PN monitoring, especially in the context of formulation adjustments in hospital settings, can be done. In the section Conclusions we added the following information: ,,In conclusion, it can be stated that the use of comprehensive information obtained from the simultaneous analysis of the element contents in serum using the TXRF technique and from statistical analysis, enables assessing the health status of the patients receiving parenteral nutrition and planning more personalized parenteral nutrition protocols. In the future, it is planned to increase the number of patients in the study and control groups and to improve the gender balance.”

Question 13. Results are heavily tabulated. Adding graphical visualizations (boxplots, forest plots, heatmaps) would improve clarity.

Authors' response: Thank to the reviewer for this note. The obtained results are presented graphically in new Figure 1.

Question 14. Ensure consistent use of abbreviations (e.g., CRP, NRS, BMI).

Authors' response: Thank to the reviewer for this note. Abbreviations are now explained when they first appear in the text.

Reviewer #3

Sampling Design and Study Cohort

Question 15. Clarify whether Measurement I represents pre-PN baseline samples or samples collected within 24 hours of PN initiation, and whether Measurement II occurred exactly seven days after initiation.

Author’s response: Thank to the reviewer for this note. Measurement I represents pre-PN baseline samples. Measurement II occurred exactly seven days after initiation. Information was included in the paper.

Question 16. Indicate if samples were collected under fasting conditions, at a standardized time of day, and under identical conditions for all participants.

Author’s response: Thank to the reviewer for this note. In the study we included the patients who received intravenous (parenteral) nutrition as their sole source of nutrients and non-nutritive components. Nutritional status and serum trace element concentrations were assessed twice, before the nutritional therapy and on the seventh day. Blood samples were collected in the morning, before the feeding bag was connected, in the surgical room of the Department of Surgery. Each patient underwent daily clinical monitoring during nutritional therapy, including fluid balance, temperature and body weight measurements, and monitoring of the central and/or peripheral venous catheter. This information was also included in the manuscript (line 182, line 195).

Question 17. Describe how control subjects were recruited and matched to patients (age, sex, and clinical status).

Author’s response: Thank to the reviewer for this note. In the paper we included the following information: ,,In the study, for comparison purposes, also the group of patients not receiving parenteral nutrition, was included. This control group consisted of 50 patients (35 women and 15 men) scheduled for planned gallbladder removal surgery (cholecystectomy), aged 30-80. Criteria for including patients within the control group were: age (from 30 to 80 years), patients admitted for planned cholecystectomy, without symptoms of cholecystitis, normal nutritional status and obtaining informed consent from the patient to participate in the study. Among women, the average age (in years) was 53 and median was 50. The youngest woman was 30, while the oldest was 80. Among men, the average age (in years) was 54 and median was 60. The youngest man was 30, while the oldest was 80 [8]. For the patients in the control group, all anthropometric, biochemical and immunological parameters and elemental analysis of the serum were measured once, before cholecystectomy.” (lines 183-194).

Question 18. Specify whether measurements were based on technical or biological replicates and whether reported values represent single determinations or averaged replicates.

Author’s response: Thank to the reviewer for this note. In TXRF serum sample analysis, for each sample the duplicate measurements were performed. Next, the average value was reported. The observed variability between measurements depends on the given element and its concentration in the sample and is on the level from about 0.1% to 20%. This information was now included in the manuscript.

Question 19. Include information on sample handling and storage (temperature, duration, number of freeze–thaw cycles) and describe how potential hemolysis was assessed or mitigated.

Author’s response: Thank to the reviewer for this note. The information was included in the text (line 211).

Analytical Methodology and TXRF Performance

Question 20. Identify any internal standard used for TXRF quantification, its concentration, and the calibration procedure.

Authors' response: Serum samples of the patients from the studied and control groups were prepared according to the following procedure: 0.8 mL of a sample was mixed with 0.05 mL of Ga (100 μg/g) used as an internal standard (line 236 in the paper). The qualitative analysis of the X-ray spectrum and the quantitative analysis of the sample elemental composition were done using spectrometer software (SPECTRA 7) based on the spectrometer calibration (line 215).

Question 21. Indicate how frequently instrument calibration was verified and whether blank or drift-control measurements were performed between sample runs.

Author’s response: Thank to the reviewer very much for this note. Now we included in the text: ,,The instrument calibration is performed each day at the beginning of the measurements. The TXRF analysis validation was performed by investigating reference control serum solution at the beginning of the studied sample measurement.” (line 215)

Question 22. Describe how data points near the detection limit were treated statistically (excluded, substituted, or handled using censored-data approaches).

Author’s response: Thank to the reviewer for this note. The data points near the detection limit were handled using censored-data statistical approach. This information was included in the manuscript (line 271).

Question 23. Clarify whether logarithmic transformations were applied prior to statistical testing and whether mean values were back-transformed for presentation.

Author’s response: Thank to the reviewer for this note. The parameters of descriptive statistics were calculated based on raw data. This information was included in the manuscript (lines 286 and 307).

The logarithmic transformations were applied for statistical testing.

Question 24. If replicate analyses were conducted, specify whether these were technical or biological replicates and report the typical within-sample variability observed.

Author’s response: Thank to the reviewer for this note. In TXRF serum sample analysis, for each sample the duplicate measurements were performed. The observed variability depends on the given element and its concentration in the sample and is on the level from about 0.1% to 20%. This information was now included in the manuscript.

Data Presentation and Statistical Reporting

Question 25. Provide a clearer overview of which elements exhibited the most notable changes between Measurements I and II (for example, Cu and Se).

Author’s response: Thank to the reviewer for this note. Increased Cu and Se concentrations after parenteral nutrition can be interpreted in the context of their specific metabolic roles, transport proteins and supplementation practices used during parenteral nutrition. Copper is added routinely to parenteral mixtures in standardized trace-element formulations. Supplementation, together with improved hepatic protein synthesis after initiation of PN, enhances the production of ceruloplasmin, which is the main plasma Cu-binding protein. Ceruloplasmin concentration responds quickly to changes in nutritional status and acute-phase reactions. A rise in ceruloplasmin increases circulating Cu, even when total body copper stores change modestly. In addition, reduced intestinal losses due to minimal enteral stimulation and improved overall metabolic balance during PN support better retention of supplemented Cu. Selenium was administered as part of standard PN supplementation. Its plasma concentration reflects selenite availability and rapid incorporation into selenoproteins such as glutathione peroxidase. PN provides controlled Se input independent of

---

## [Decision Letter · Decision Letter 1]

2 Jan 2026

Study of element concentrations in blood serum of patients receiving parenteral nutrition using total reflection X-ray fluorescence analysis

PONE-D-25-39141R1

Dear Dr. Aldona,

We’re pleased to inform you that your manuscript has been judged scientifically suitable for publication and will be formally accepted for publication once it meets all outstanding technical requirements.

Kind regards,

Satish Rojekar, Ph.D.

Academic Editor

PLOS One

Additional Editor Comments (optional):

Reviewers' comments:

Reviewer's Responses to Questions

**Comments to the Author**

Reviewer #1: All comments have been addressed

Reviewer #2: All comments have been addressed

Reviewer #3: All comments have been addressed

Reviewer #4: All comments have been addressed

2. Is the manuscript technically sound, and do the data support the conclusions?

Reviewer #1: (No Response)

Reviewer #2: Yes

Reviewer #3: Yes

Reviewer #4: Yes

3. Has the statistical analysis been performed appropriately and rigorously?

Reviewer #1: (No Response)

Reviewer #2: Yes

Reviewer #3: Yes

Reviewer #4: Yes

4. Have the authors made all data underlying the findings in their manuscript fully available?

Reviewer #1: (No Response)

Reviewer #2: Yes

Reviewer #3: Yes

Reviewer #4: Yes

5. Is the manuscript presented in an intelligible fashion and written in standard English?

Reviewer #1: (No Response)

Reviewer #2: Yes

Reviewer #3: Yes

Reviewer #4: Yes

Reviewer #1: (No Response)

Reviewer #2: All reviewer comments have been satisfactorily addressed, resulting in an improved manuscript. I recommend acceptance in its current form.

Reviewer #3: Thank you for your thorough revisions. The manuscript has improved substantially in methodological clarity, data transparency, and clinical interpretation.

The clarifications regarding sampling design, timing of Measurements I and II, fasting conditions, and control-group recruitment now provide a solid foundation for understanding the study cohort. Details added on sample handling, TXRF methodology (internal standard, calibration frequency, duplicate measurements), and treatment of values near detection limits greatly strengthen the analytical rigor.

pasted The explanation of statistical procedures, use of raw values for descriptive statistics, log-transformation for hypothesis testing, paired analyses for within-subject comparisons, now makes the data analysis more transparent. The discussion of variability in Fe, Zn, and Cr and the relevance of log-stable distributions is helpful. The expanded biological interpretation of Cu, Se, Zn, and Fe behavior during PN provides meaningful clinical context. The comments on element correlations (e.g., Ca–S, Cr–Ca) and their possible metabolic or PN-formulation origins are logical and informative. The identification of Cu and Se as the most clinically informative markers for routine monitoring adds translational value. Study limitations and future directions are now appropriately acknowledged, and editorial corrections have improved readability.

Overall, the revisions satisfactorily address the reviewer questions, and the manuscript is now clearer, more rigorous, and more clinically relevant.

Reviewer #4: (No Response)

**Do you want your identity to be public for this peer review?** For information about this choice, including consent withdrawal, please see our Privacy Policy

Reviewer #1: **Yes:** Isha Dhami

Reviewer #2: **Yes:** Deepika Godugu

Reviewer #3: **Yes:** RASHI PORWAL

Reviewer #4: No

---

## [Editor Report · Acceptance letter]

PONE-D-25-39141R1

PLOS One

Dear Dr. Kubala-Kukuś,

I'm pleased to inform you that your manuscript has been deemed suitable for publication in PLOS One. Congratulations! Your manuscript is now being handed over to our production team.

Kind regards,

on behalf of

Dr. Satish Rojekar

Academic Editor

PLOS One